# Women's experiences of the Odon Device to assist vaginal birth and participation in intrapartum research: a qualitative study in a maternity unit in the Southwest of England

Emily J Hotton  ,[1,2] Natalie S Blencowe  ,[3,4] Erik Lenguerrand,[1,2] Tim J Draycott,[2] Joanna F Crofts,[2] Julia Wade  [5]

¹Translational Health Sciences, University of Bristol, Bristol, UK
²Women's and Children's Research, North Bristol NHS Trust, Bristol, UK
³Centre for Surgical Research, School of Social and Community Medicine, University of Bristol, Bristol, UK
⁴University Hospitals Bristol NHS Foundation Trust, Bristol, UK
⁵Population Health Sciences, Univeristy of Bristol, Bristol, UK

**Correspondence to**
Dr Emily J Hotton;
emily.hotton@nhs.net

## ABSTRACT

**Objective** To investigate women's experiences of having a birth assisted by the Odon Device (an innovative device for assisted vaginal birth) and participation in intrapartum research.

**Design** Qualitative semistructured interviews and observations undertaken in the context of case study work embedded in the ASSIST feasibility study.

**Setting** A tertiary referral National Health Service (NHS) maternity unit in the Southwest of England, between 8 October 2018 and 26 January 2019.

**Participants** Eight women, four operators and 11 midwives participated with eight observations of the assisted vaginal birth, eight interviews with women in the postnatal period, 39 interviews/reflections with operators and 19 interviews with midwives. Women in the case study research were recruited from participants in the main ASSIST Study.

**Intervention** The Odon Device, an innovative device for assisted vaginal birth.

**Results** Thirty-nine case studies were undertaken. Triangulation of data sources (participant observation, interviews with women, operators and midwives) enabled the exploration of women's experiences of the Odon Device and recruitment in the intrapartum trial. Experiences were overwhelmingly positive. Women were motivated to take part by a wish for a kinder birth, and because they perceived both the recruitment and research processes (including observation) to be highly acceptable, regardless of whether the Odon-assisted birth was successful or not.

**Conclusions** Interviews and observations from multiple stakeholders enabled insight into women's experiences of an innovative device for assisted vaginal birth. Applying these qualitative methods more broadly may illuminate perspectives of key stakeholders in future intrapartum intervention research and beyond.

**Trial registration number** ISRCTN10203171; ASSIST Study registration; https://doi.org/10.1186/ISRCTN10203171.

## INTRODUCTION

Globally, 140 million babies are born each year, with an estimated 2.6 million stillbirths

### Strengths and limitations of this study

► We used qualitative methodology to assess the experiences of women undergoing attempted assisted vaginal birth using the Odon Device as part of a clinical trial.

► We obtained observation and interview data from women, operators and midwives to capture a broad range of experiences.

► Our sample sizes were appropriate to provide initial insights; however, they may be limited in capturing the full nuance of opinions and attitudes.

(50% related to intrapartum complications),[1] 2.5 million neonatal deaths (24% related to intrapartum complications)[2] and 300 000 maternal deaths (4%–13% related to a complicated or prolonged second stage of labour).[3] It is suggested that a significant proportion of births complicated by a prolonged second stage of labour could be resolved by performing an assisted vaginal birth (AVB). However, despite AVB being recognised by the WHO as one of the seven critical components of basic emergency obstetric and newborn care,[4] the global rate of AVB has continued to decline significantly. Declining rates of AVB have been attributed to concerns regarding: complications of AVB and the associated litigation,[5] limited availability of devices in certain settings, public perception[6] and insufficient training in device use or complex interventions.[7]

AVB is associated with better maternal and neonatal outcomes than Caesarean section at full dilatation, when performed by a skilled operator in an appropriate setting.[8] Specifically, women having an AVB have reduced rate of postpartum haemorrhage and reduced

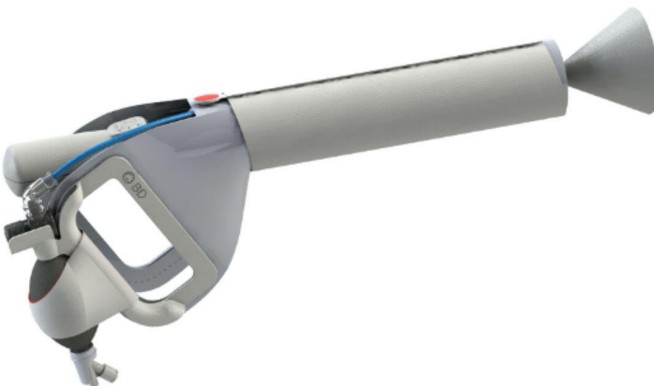

**Figure 1** The Odon Device. Courtesy and Becton, Dickinson and Company reprinted with permission.

length of stay, and babies are less likely to be admitted to the neonatal intensive care unit (NICU).[8]

AVB is almost always performed in an emergency setting, limiting time for information provision.[9] Nevertheless, it is important to ensure that women and their birth partners, feel informed, empowered and supported during their birth experience.[10 11] A recent review suggests that AVBs can be seen as positive by women if they receive good communication from the team and feel involved in decision-making.[12] This is despite the general negative public perception of AVB, particularly forceps.

The consequences of a poorly performed AVB can be devastating and long lasting for mother, baby and operator. Comprehensive training, appropriate supervision and support can drastically improve outcomes and patient and operator experience.[13–16]

Despite recognition of AVB as a life-saving intervention, there have been no significant developments in AVB devices since the 1950s when the ventouse/suction cup was introduced.[17] The Odon Device (figure 1) was invented by an Argentinean Car Mechanic and then further developed by a multiprofessional team of doctors, midwives and engineers as an alternative device.[18–22] The device consists of a plastic applicator and polyethylene sleeve. The sleeve is placed over the fetal head and contains a circumferential air chamber that is inflated, providing purchase for the operator to apply traction.[22]

A core component in any evaluation of a novel device for AVB is the investigation of women's experiences. Qualitative case study methodology has previously been employed in maternity settings,[23] exploring a range of topics from care pathways for women giving birth in Birth Centres[24] to the psychosocial impact of high-risk pregnancy.[25] However, there are no published examples of case study methodology being used in intrapartum studies of clinical interventions. There is limited research exploring the experiences of women regarding recruitment processes during intrapartum research[26]; those that exist report that gaining consent for intrapartum research can be challenging, especially if women are in pain or fatigued.[27]

This study investigated women's experiences of the Odon Device and participation in intrapartum research through exploring women's and health professional views and combining them with observations collected as part of a broader case study investigation embedded within a single arm feasibility study of the Odon Device for clinically indicated AVB—the ASSIST Study.[22 28]

## METHODS
### Research design
Qualitative case study methodology enables systematic study of phenomena across cases, enabling identification of similarities and differences and resulting in detailed, contextualised insights. The case does not have to be a person but could be an event, decision, action or object.[29] In this study, a case was the event surrounding the single use of the Odon Device. This methodology is particularly useful when exploring the introduction of the Odon Device as the context includes multiple variables: a range of reasons for requiring AVB, women, allied healthcare professionals and operator.

### Study context
The study was undertaken in a large tertiary referral National Health Service (NHS) hospital in the Southwest of England with approximately 6000 births per year. It has an alongside NICU that accepts babies from across the region. The Maternity Department supports women, with extensive options for birthplace including: homebirth, midwife-led independent birth unit, alongside midwifery unit and an obstetric-led birth unit for high-risk cases. Approximately, 11% of all births in the maternity unit were AVBs. These were all conducted on the obstetric-led birth unit by an obstetrician with the support of a midwife.

### Participants
Women were eligible to take part in ASSIST if they were: aged 18 years or older, may require an AVB with a live, singleton pregnancy with negative antenatal screening to blood-bourne viruses.[22] Women were recruited to the ASSIST study (in the antenatal period or during the latent phase of labour) by a research midwife and were given a participant information leaflet, bespoke information video and had a detailed discussion with the recruiting midwife.[28] Written consent for the integrated qualitative research reported here was provided by women concurrently at the time of providing consent to participate in the ASSIST Study (either antenatally or in the latent phase of labour). All Odon Device operators and midwives provided written consent to participate.[28]

### Sampling
Typical sampling, a form of purposive sampling appropriate for case study methodology, was used to include a range of indications for AVB and a range of operators allowing for the understanding of multiple perspectives

**Table 1** Demographics and characteristics of women participating in the case study research compared with those in the ASSIST study

| | Case studies n=8 (%) | ASSIST Study n=40 (%) |
|---|---|---|
| **Parity** | | |
| Nulliparous | 7 (87.5) | 34 (85.0) |
| Multiparous | 1 (12.5) | 6 (15.0) |
| Mean age in years (mean) | 28.5 | 29 |
| **Ethnicity** | | |
| White British | 8 (100.0) | 32 (80.0) |
| Any other white background | 0 (0.0) | 4 (10.0) |
| Black | 0 (0.0) | 2 (5.0) |
| Indian | 0 (0.0) | 1 (2.5) |
| Any other Asian background | 0 (0.0) | 1 (2.5) |
| **Mode of labour onset** | | |
| Spontaneous | 0 (0.0) | 6 (15.0) |
| Induced | 8 (100.0) | 34 (85.0) |
| **Analgesia at time of birth** | | |
| Epidural | 4 (50.0) | 34 (85.0) |
| Perineal infiltration | 3 (37.5) | 12 (30.0) |
| None | 1 (12.5) | 1 (2.5) |
| **Final mode of birth** | | |
| Caesarean section | 0 (0.0) | 2 (5.0) |
| Successful birth with Odon Device | 5 (62.5) | 19 (48.0) |
| Assisted vaginal birth with forceps or ventouse (unsuccessful Odon Device) | 3 (37.5) | 19 (48.0) |

across a range of variables and for results to be more transferable.[30 31]

## Patient and public involvement

Patient and public involvement (PPI) was undertaken for all aspects of the ASSIST Study.[22] A session on the qualitative aspects of the study was undertaken 22 March 2018. Twelve people (pregnant women n=8 and their birth partners n=4) took part. Participants were provided with the participant information leaflet, consent form and interview topic guide to review prior to the session. Opinions on being observed during labour and interviewed postbirth were specifically sought. There was unanimous agreement that the qualitative research would be accepted and welcomed by women. The opportunity to speak with an obstetrician about the birth on the first postnatal day was viewed as a positive opportunity. No substantial changes to qualitative design of the study were made as a result of this session.

## Data collection

The case study investigation included data from one or more of the following sources for each case: participant observation of the attempted Odon-assisted delivery and interviews with women, operators and midwives. Conducting case studies in the clinical context of device use, with triangulation of data from different sources, generates greater depth of insight through the process of data triangulation, than would be available using other approaches such as serial interviews with women and operators.[32]

Participant observations commenced when the obstetrician entered the woman's room to conduct the AVB and ended when the obstetrician left the room following the birth of the baby. The researcher was a clinical obstetrician and part of the ASSIST Study Team (EJH). Contextual factors, verbal and non-verbal communication were documented on an observation schedule. The observation schedule was developed during preparatory work, prior to the commencement of the ASSIST Study (online supplemental file 1). Observations ceased when no new insights were derived from two consecutive observations. All women who had an observed AVB with the Odon Device were invited to participate in an interview on the first postnatal day. The operator and midwife present at a birth using the Odon Device were invited to take part in separate interviews. Interviews were semistructured and directed by topic guides (online supplemental file 2), which were established based on preclinical and clinical literature and developed iteratively during data collection. Each operator and/or midwife could be interviewed more than once during the course of the study as each case was defined by the use of the Odon Device and not the clinicians present. Where possible, all interviews were undertaken face-to-face.

## Triangulation of data

Transcripts were created from the participant observations made during the birth. All thoughts and interpretations of the researcher were noted, often aiding the creation of specific interview questions. Any issues that required clarification from participants were highlighted during the transcription process for further discussion in subsequent interviews. Findings were successively tested and refined by the researcher as additional transcripts were analysed and explored in the interviews with participants.

## Data analysis

Data collection and analysis ran iteratively and in parallel.[33] NVivo V.12 (QSR International, Melbourne, Australia) was used to organise data from all sources as well as supporting analysis. Thematic analysis was used for analysis, using the key steps that have been well described in the literature.[33] A broadly inductive approach to analysis was used, enabling themes to be derived from the data. Triangulation of sources guided the analysis of observation data and interview transcripts from a single case to identify, outline and organise codes. Codes were then grouped together where commonalities were noted to identify initial themes; the description and sample quotes to illustrate the theme were noted. Finally,

case summaries were developed and reviewed to create thematic accounts, which identified patterns within the data and provided explanations for these.[34] Double coding of a proportion (20%) of interview transcripts was undertaken by JW. Commonality and variances in coding between qualitative researchers were discussed, resolved by consensus and used to further shape developing themes and sampling.

## Reflexivity

All case study research, PPI, piloting, data collection and analysis were undertaken by a medically qualified obstetrics and gynaecology trainee, Odon Device trainer and Odon Device operator for the ASSIST Study (EJH). The use of the Odon Device to assist birth is highly complex and often takes only minutes to perform; without an in-depth knowledge of the device, its instructions for use and AVB, subtle differences or anomalies in its use may have overlooked during data collection by a non-clinical professional. It is possible that preconceived ideas and suppositions were brought to the study given the researcher's specialist clinical background. To maintain rigour, reflexive comments were made after each data collection event (observation or interview) and during data analysis. These were discussed with an experienced independent non-clinical qualitative researcher (JW).

## RESULTS

Presented here is the combination of observational and interview data relevant to women's experiences of the attempted Odon-assisted delivery and participation in the intrapartum study. Other data relating to optimising use of the device by operators have been analysed separately and will be published separately.

The 39 case studies conducted of attempted Odon-assisted births, included eight observations, eight interviews with women observed, 19 midwife interviews, 37 operator interviews and two operator reflections. Demographics and characteristics of the women included are presented in table 1 and data were collected for the 39 case studies in table 2.

Observations varied in length from 33 min to 68 min. Interviews with women lasted between 7 and 10 min, interviews with midwives lasted between 3 and 13 min and interviews with operators lasted between 5 min and 26 min.

A summary of the themes relating to the women's experiences is presented in figure 2.

## Theme 1: reasons for taking part—a different assisted birth

All eight women interviewed gave very similar reasons for agreeing to participate in the ASSIST Study and have the Odon Device used to assist the birth of their baby. Women's experiences were predominantly positive even where Odon use had not succeeded.

Half the women interviewed (n=4) stated a specific aversion towards forceps-assisted birth as an important factor

**Table 2** General information about the case studies (n=39)

| Case study number | Observation | Women | Operator | Midwife |
|---|---|---|---|---|
| | | **Data collected** | | |
| | | **Interviews** | | |
| 1 | X O01 | X W01 | X D02 | X M15 |
| 2 | | | X D03 | X M18 |
| 3 | | | X D03 | X M12 X M18 |
| 4 | | | X D03 | X M10 |
| 5 | | | X D02 | X M18 |
| 6 | X O02 | X W02 | X D02 | X M19 |
| 7 | X O03 | X W03 | X D03 | X M11 X M11 |
| 8 | X O04 | X W04 | X D02 | |
| 9 | | | X D03 | X M16 |
| 10 | X O05 | X W05 | X D02 | X M13 |
| 11 | X O06 | X W06 | X D02 | X M14 |
| 12 | X O07 | X W07 | X D04 | X M12 |
| 13 | | | X D02 | X M16 |
| 14 | | | X D03 | X M10 |
| 15 | X O08 | X W08 | X D03 | X M17 |
| 16 | | | X D02 | |
| 17 | | | X D04 | X M20 |
| 18 | | | X D06 | |
| 19 | | | X D06 | |
| 20 | | | X D04 | X M11 |
| 21 | | | X D06 | |
| 22 | | | X D02 | X M11 |
| 23 | | | X D02 | |
| 24 | | | X D06 | |
| 25 | | | X D06 | |
| 26 | | | X D02 | |
| 27 | | | X D02 | |
| 28 | | | X R01 | |
| 29 | | | X D02 | |
| 30 | | | X D06 | |
| 31 | | | X D02 | |
| 32 | | | X R02 | |
| 33 | | | X D03 | |
| 34 | | | X D03 | |
| 35 | | | X D06 | |
| 36 | | | X D02 | |
| 37 | | | X D06 | |
| 38 | | | X D06 | X M11 |
| 39 | | | X D04 | |

in their decision-making. Midwives shared the antipathy for forceps and the wish for an alternative device and an AVB that women would welcome. Aversion towards forceps was linked to previous negative experiences of friends or family:

**Figure 2** Women's experiences: women's and health professional views—themes and sub-themes.

W04: And also the forceps, to me, looked like a torture weapon, so I was keen to give something else a go.

W01: I don't have a lot of experience of it, but I know my nephew had forceps and was very bruised and marked.

Women and midwives articulated similar views, the lay belief that forceps are more 'traumatic' either physically or emotionally:

M14: …but it definitely didn't look quite as traumatic as a forceps delivery… But I do feel like, for her, it was a nicer experience than it might have been for a forceps. I don't know if it's the word 'forceps' that traumatises ladies, but everyone seems to be terrified of them.

Midwives perceived that the Odon Device was kinder than other devices and described the births as 'softer', 'nicer', 'beautiful', 'easy', 'smooth' and 'gentle':

M18: The device use and the delivery was actually kind of one of the most beautiful deliveries of an assisted device used that I've seen.

This belief was also reinforced by observations from women, midwives and operators, which babies successfully delivered with the Odon Device had minimal scalp or face injury following birth. The reduction in visual marks on the baby seemed to have a big impact on the overall perception of the birth:

M20: These babies are coming out with no marks on them, it's fab. I think it's stunning. It's absolutely really good.

W07: Over the moon. The fact that it has helped with his delivery, the fact that he's safe, there's no marks on him whatsoever, he's completely fine, and he's just a happy baby. So, that's all I could ever ask for really.

Observations highlighted that all staff members present at time of birth, both clinical and research, strove to ensure that the birth experience was positive for women, perhaps more so than in usual clinical practice. Environmental factors seemed to stand out as creating a calm birthing experience:

O01: Low lighting in the room, not main lights but two spotlights providing the light to the operator only. Music on in the background was on already prior to entering the room. It was a very calm and quiet environment.

Team members perceived that birth experience was more positive for women who were felt to have really wanted a birth assisted by the Odon Device:

M11: And then at that point she said, "Oh, come on doctor," she said something like, "I've got faith in you, you can do it." So, she [the patient] really, really wanted him [the operator] to do it. And then she [the patient] said, "But if this doesn't work, I don't want forceps because of the baby's face; I'd rather have a section."

The level of engagement of women and their birth partners with the Odon Device surprised the operators:

D02: … the husband was leaning over her leg and watching exactly what I was doing and was commentating as I was applying the device… So, when I got to four, he was like, "It's on four, it's on four, it's nearly there, nearly there." I wasn't quite sure whether he was encouraging his wife or encouraging me, but it made me smile.

Initial generalised altruism regarding research participation was overtaken by genuine engagement and specific interest in the study:

W01: I'm quite cynical and I was like, "I'm not going to hear about this again. I just have to say yes to this." Then I read about it and actually I did find it quite interesting, so I was more than happy to sign up to it.

Several women stated their surprise when they needed an AVB having assumed during the consenting process that they would not require an assisted birth.

W06: It's such a low percentage, isn't it, probability-wise. So, you just, kind of, almost think, "Well, it probably won't come to this. So, might as well [consent]."

Women expected and presumed that there would be observations of device use in the ASSIST Study. They appreciated it was important to observe the device in real-life settings and to learn from its use:

W02: You're not going to know any different. It's easy saying, "simulations, we'll do these," but until it's gone into practice on a real baby…You need to observe to see how it works…Or how can you, maybe, if something went wrong, fix it. How are you ever going to learn if anything…

When asked about the possibility of videoing the AVB instead of having a formal observation by a researcher, views from women were very mixed. For some it was acceptable, providing there was anonymity and that it was used for training, one stated 'no'.

### Theme 2: What makes an acceptable recruitment process?
Women valued various aspect of the recruitment process, including provision of information through multimedia format, the manner of the recruiter, timing of when they were approached and treating consent as a process and not a single event, so checking consent multiple times.

The provision of the information video allowed women to visualise and understand how the Odon Device works. They already knew what was going to happen at the time of assisted birth:

W02: …we watched the video. That's what, I think, made us say yes, I suppose, the video. Just, it's quite a good little demo on the video that makes you think…

Women preferred this innovative multimedia approach rather than reading the information leaflets:

W04: A leaflet you tend not to read it, do you? And a presentation is dull. So yes, I quite liked the video.

They appreciated the opportunity to see the team and operators in the participant video.

The way recruitment was delivered by the research team was also of the upmost important to women. The women described how the manner of checking consent and regular interactions with the research team increased the trust in them:

W06: The woman that came round, she had the right approach. You don't want to be too pushy or whatever, and you need to, yes, have tact with people, and she was fine. She was good.

Research follow-up was experienced as supportive for the women:

W07: Even after delivery, just coming in to check and see that everyone's alright, honestly, I didn't find any faults.

### Theme 3: acceptability of the research process
For the women interviewed, birth experiences were not affected by whether the Odon Device was successful or not or the number of healthcare professionals present at the time of birth.

W02: I know, obviously, it wasn't successful, but it's not going to be successful every single time. Everyone's situations are different (unsuccessful Odon birth).

Interestingly, women focused on outcomes for their baby and their feelings at the time of birth rather than their own pregnancy outcomes or birth complications when discussing their experiences.

Women could feel overwhelmed at the time of needing an AVB:

O06: At the time of needing an AVB, she was losing the plot and scared, she said "help me" "I cannot do this" (successful Odon birth).

Despite this women reported feeling cared for and the situation felt routine. When asked about their birth experience, women reflected on the feeling they had during the birth of their baby:

W07: But the device and that, I didn't even make… I know, obviously, it's a big thing, but I didn't even really pay any attention to it. I just thought it was a normal device, like forceps (successful Odon birth).

Some women ascribed an unsuccessful Odon birth to their baby rather than to device failure. Interestingly, they saw an unsuccessful AVB with the Odon Device as providing useful insights for future understanding and births:

W02: I know, obviously, it wasn't successful, but it's not going to be successful every single time. Everyone's situations are different. It was just our baby being

difficult. It's different if every single person going didn't like it. But no, I'd, 100%, consent again (unsuccessful Odon birth).

Some women were surprised that there were increased numbers of staff present at the time of birth but did not state that this affected their birth experience:

W04: I must say, it was quite overwhelming when everyone came in at the time the help was needed… They explained they were all for different stuff and for different jobs (successful Odon birth).

Half of the women stated that having extra staff present at time of birth was fine as they perceived that the more people, the safer the environment:

W02: I think the more people there, the less you can worry, because you know you're in safe hands…Yes, but you're thinking, "Right, there are 10 people in this room now, so someone's going to be alright" (unsuccessful Odon birth).

## DISCUSSION
### Main findings

Qualitative research in the form of a case study investigation into the feasibility of using an innovative device for AVB was used in this study to investigate women's experiences of the attempted Odon-assisted birth and of participating in intrapartum research. This study included observations of attempted AVBs, interviews with women involved and interviews with midwives and operators to explore women's experiences in detail. Women's experiences were overwhelmingly positive (more so than had been expected by the study team), regardless of the outcome of using the device. Study recruitment and follow-up processes were valued by women, and in-person observation of the birth was acceptable.

Women and midwives want an alternative to forceps and ventouse. There are wide inconsistencies in the use of AVB worldwide and these are often multifactorial, relating to concerns regarding complications of AVB and the associated litigation,[5] limited availability of devices in certain settings, public perception[6] and insufficient training in devices or complex interventions.[7 12] Our study highlighted a clear maternal aversion towards forceps and a wish for a kinder AVB. There was genuine engagement in research and positive feedback from midwives regarding their perception of the device. Women appreciated the recruitment process and the methodology used for information provision, especially via the participant information video. Providing the women with an information video proved vital in aiding their understanding of the innovative device as well as being a welcome format for receiving information. Women understood the need for qualitative research and expressed an awareness of the need to observe and learn from each assisted birth. Unsuccessful births with the Odon Device were not viewed by women as a failure, rather a learning opportunity for the research team. There was some reservation from midwives regarding the number of people present in the room for an AVB with the Odon Device, especially with formal observation; however, it was viewed as acceptable if women were happy. These findings are very similar to a recent systematic review on women's experiences of AVB.[12] As it was the first time the Odon Device had been used in clinically indicated cases,[22] there was acceptance and expectation from women that their births be formally observed. There was less evidence that video recording the births (not undertaken in this study) was acceptable to women.

### Strengths and limitations

This is the first study to undertake an in-depth qualitative exploration of the Odon Device for clinically indicated AVBs. Capturing the experiences of women is vital in assessing the device in clinical trials. Conducting case study methodology on the labour ward enabled researchers to gather a 'genuine' representation of experience within the environment, in which the device is used that might not have been achieved using alternative approaches.[32] Another key advantage of gathering data from multiple sources (participant observation and interviews) is the opportunity for triangulation of a wealth of data. Data amalgamation (within cases) combined with constant cross-case comparison improved the validity of the study findings. A further strength is that where there are indications that saturation has been reached, there is an opportunity for one form of data collection to be dropped, applied here to observation data. Reporting was undertaken following the Standards for Reporting Qualitative Research[34] (online supplemental file 3).

Despite these advantages, there are limitations to this study. Only women accepting participation in ASSIST were observed or interviewed and those declining participation may have different views. There were only eight case studies involving observation and interviews with women, because data saturation was reached from observations. This limitation raises the possibility that findings are less generalisable. Furthermore, the interviews with women were short. There are several reasons as to why this may be the case. First, women were only asked a very small number of questions specifically relating to their birth experiences. Operators and midwives, however, were asked questions not only on experience but also on device use, technique, design and trial methodology. Additionally, women were interviewed at day 1 postnatal, when likely fatigued. This was discussed during the PPI sessions and women agreed that interviewing women in hospital close to the birth was more suitable than later at home, as women were aware that their memory of their birth experience may not be reliable. The case studies were undertaken by a specialist trainee in obstetrics and gynaecology meaning that preconceptions and existing knowledge may have influenced the collection and interpretation of the data. However, at the time of commencing

the case studies, the researcher was naïve to the use of the Odon Device in the clinical setting. Finally, operators may have changed their behaviours during observations, perhaps not reflecting their real-life practice. It is possible that there was response bias as women knew that the researcher was a clinician in the trial. However, this was discussed during PPI and valued as an attraction of the study for women. Women and their partners were keen to discuss their birth with a clinician who knew the device, feeling it was an essential aspect of the research process.

## Interpretation

With novel devices being tested in clinical trials, it is vital that experiences of key stakeholders are explored. Having an innovative device for AVB that has high clinical efficacy is not useful if women find it unacceptable. Our findings suggest that women are willing to accept an alternative device for AVB, especially one that is perceived as kinder to their babies. Women commended the use of video during the recruitment process and expressed support for intrapartum research.

## CONCLUSION

This study demonstrates that qualitative case study methodology enables the detailed exploration of experiences and views of women and healthcare professionals in intrapartum research and that the methods (especially observation) were acceptable and expected by women.

When introducing or implementing a novel device into clinical use, it is vital that the experiences and opinions of stakeholders are considered. Case studies provide a way of exploring these views in detail by integrating data from multiple sources, providing a comprehensive and authentic exploration. Case study methodology is used widely in other avenues of research; however, it has not yet been widely adopted in implementation research during device or surgical trials.[35] Successful completion of case studies on labour ward for an intrapartum intervention has provided methodology that can be transferred to intervention delivery and design of novel devices. Women appear keen to support the investigation of novel devices for AVB, such as the Odon Device, and value the importance of information sharing during the recruitment process. The perception of women can help shape and design future recruitment strategies, information provision and intervention delivery. The use of case study methodology must be further tested in alternative settings and specialties.

**Acknowledgements** The authors would like to thank all the women who agreed to take part in this research and all the maternity staff who enabled to safe completion of the ASSIST Study.

**Contributors** EJH, NSB and JW developed the concept for the qualitative research. EJH performed all data collection and analysis with co-coding performed by JW. EJH wrote the initial draft of the manuscript with support from NSB, JFC and JW. EJH, NSB, EL, TJD, JFC and JW reviewed and approved the final manuscript. JFC is guarantor of the study.

**Funding** This research was supported by the Bill & Melinda Gates Foundation [OPP1184825/INV-010180]. The Foundation had no role in study design, planning, conduct, analysis or publication production.

**Competing interests** EL is a member of University of Bristol (UoB) and part of his salary is paid by PROMPT Maternity Foundation (PMF); PMF has received funding from a Saving Lives at Birth award via a subcontract from Becton, Dickinson and Company (BD) to conduct preclinical simulation studies of the Odon Device, these funds have been used towards the salary of TJD and JFC when undertaking the simulations studies. All other authors report no competing interests.

**Patient consent for publication** Consent obtained directly from patient(s)

**Ethics approval** This study involves human participants and was approved by The research was approved by South Central–Berkshire REC, UK on 3 September 2018 (18/SC/0344), the MHRA on 9 August 2018 and the HRA on 3 September 2018. Participants gave informed consent to participate in the study before taking part.

**Provenance and peer review** Not commissioned; externally peer reviewed.

**Data availability statement** Data are available upon reasonable request.

**ORCID iDs**
Emily J Hotton http://orcid.org/0000-0002-8570-9136
Natalie S Blencowe http://orcid.org/0000-0002-6111-2175
Julia Wade http://orcid.org/0000-0001-6486-6477

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
