## [Reviewer comments · BMJ Open]

ARTICLE DETAILS

TITLE (PROVISIONAL)	Women's experiences of the Odon Device to assist vaginal birth and participation in intrapartum research: a qualitative study in a maternity unit in the Southwest of England
AUTHORS	Hotton, Emily; Blencowe, Natalie; Lenguerrand, Erik; Draycott, Tim J; Crofts, Joanna; Wade, Julia

VERSION 1 – REVIEW

REVIEWER	Claire Feeley University of Central Lancashire
REVIEW RETURNED	29-Sep-2021

GENERAL COMMENTS	Many thanks for giving me this opportunity to review this paper that focuses on women's experiences of the Odon device when an assisted birth is required. It is a timely topic as there are growing calls to improve the use of instruments during birth, so the innovation of a new device that includes the voices of the women's experiences (and reports from the health professionals) is a vital contribution. However, the stated objectives to the findings presented are somewhat disjointed. Most of the findings related to the recruitment processes, motivations and experience of participating in such a trial. This is really important and valuable information but would need substantial revision to ensure there is consistency throughout the paper. For example, the first half of the paper indicates insights about the experience of an Odon device birth, yet only one theme that appears to be abbreviated into a Table reflects those experiences. The rest of the findings, discussion and conclusion relates to the successful PPI engagement and recruitment strategies rather than the birth experience itself. As stated, I feel this is a tremendous success but the paper needs revising accordingly to ensure the key messages are consistent and maintained throughout the paper. Abstract: Clear, concise and well written (however, will need amending following revisions). Introduction: Clear and well written but with a few revisions required: 1. I suggest adding a bit more contextual info i.e. reasons as to why AVB use has declined, the concerns around the other instruments (morbidity) etc, to further contextualise/funnel into the information about the Odon device.2. I'm also not convinced such a lengthy point about the justification for the case study approach is needed in the Intro, rather integrate the justifications into the methods under a sub-heading 'research design' and use the word count to unpack the issues around AVB more.3. Given my comments under results, this may be better revised to include information of challenges/successes of intervention trials
--

	particularly in maternity care, particularly during 2nd stage of labour Methods: Mostly well written but suggest:  1. Re-ordering for a better flow and with use of more subheadings to guide the reader (I had to go back and forth to fully follow this section) e.g. Research design, study context (hospital info), recruitment & sampling and so forth. Also type of data collected be moved to the 'data collection' section. And would suggest the PPI section comes before data collection/analysis, otherwise leaves the reader wondering about the ethical considerations of the study. 2. Would suggest giving some examples of the interview questions or at least the topics covered as there is only one mention (abstract) that the study was to explore motivation/participation in the study itself in addition to the experience of an Odon device birth. 3. Data analysis needs some further explaining and justification as to the methods used. Which method was followed and according to what framework/guidance/process? The references include Braun & Clarke which indicates thematic analysis and then Brink et al for grounded theory, be good to understand the rationale. If methods are combined or a novel approach has been taken, then this needs to be explicit. Great to see a reflexivity statement! Results  1. Info about data saturation needs to be in methods (and again suggests GT approach not TA, as data saturation is highly contested in other methodologies, needs revising please). Keep results to what you have found, again, makes it easier for the reader to follow and enhances clarity. 2. Info about the observations/interviews (see lines 218-222) also would read better in the methods section for the same reasons. 3. The themes address women's motivation to participate and their experiences, but it appears 3 out of 4 themes are about recruitment/motivation to participate, this needs clarification throughout the paper from the abstract on and consider adapting the title and amending the figure. They are really important findings and great contribution to the field, but it reads as disjointed to the stated objectives where I was expecting much more on the experience of an Odon birth. 3. Theme 2, the quote used at first glance reads as coercive, I had to read it a couple of times to realise the point, I would suggest changing the title or using a different theme name (a quote wasn't used in Theme 1). It also suddenly introduces key information about the recruitment strategy not evident in the methods. The use of videos is great but is essential to know about it in the methods. 4. Theme 3 is underdeveloped and doesn't provide very rich insights to the experiences. The use of a table without the narrative doesn't sit in line with the other themes. 5. Theme 4 doesn't have enough insights to be a theme, would suggest integrating it into another. Discussion: Given my other comments I would suggest revising this section, needs greater exploration of the wider literature including women's views/experiences of AVB to support this section. Nor does it link to operators views and experiences which would strengthen the points made here. Conclusion: Reads well and marries up to my comments about the key point of this paper which appears to about successful PPI and recruitment engagement to a challenging field for intervention trials.
--	---

	If the manuscript is revised with this conclusion in mind, it will provide a structured 'golden thread' with valuable information for other healthcare professionals and trialists.
--	---

REVIEWER	Hong Jiang Fudan University
REVIEW RETURNED	03-Oct-2021

GENERAL COMMENTS	The study aimed to investigate experiences of women who gave birth assisted by the Odon Device, an innovative device for assisted vaginal birth via observation and individual interviews with women, women's partners, operators, and midwives. The research topic is interesting and the manuscript provides some useful information for the practice of Odon Device for assisted childbirth. However more details of the reflections and perspectives about the experience of device usage are expected and it will contribute to the application of Odon Device in obstetric practice and the improvement of clinical service quality. Please consider the following questions for further improving the manuscript.  1. Authors described this study used a case study research method. However, the observation and individual interviews are different kinds of qualitative study. The research methods used in this study, including sampling, data collection and data analysis, seem more like qualitative research methods. Please reconsider the methodology of this study. 2. It is expected to introduce the Odon Device in Background in more details such as the conditions of application, approaches of its usage, and the innovation and difference compared to other devices. 3. For qualitative data analysis, it is usually required two independent researchers to carry out coding. This study reported 'Double coding of a proportion of interview transcripts was undertaken by JW'. Please provide the specific proportion in this section. 4. Table 2 seems not necessary and its contents can be described briefly. A table introducing the characteristics of participants might be needed. 5. Most of the first three paragraphs of Result are about data collection. It is supposed to be in Methodology part. 6. For some quotes, it is a bit hard to catch. For example, in P16, -- M11: 'And then at that point she said, "Oh, come on doctor," she said something like, "I've got faith in you, you can do it." So, she really, really wanted him to do it. And then she said, "But if this doesn't work, I don't want forceps because of the baby's face; I'd rather have a Section."' It is confusing whom 'she' and 'him' referred to. Maybe 'She' was woman, and 'him' was the doctor. 7. P19, it is suggested to use "pregnancy outcomes or complications" rather than "birth outcomes or complications" in the sentence "women focused on outcomes for their baby and their feelings at the time of birth rather than their own birth outcomes or complications." 8. The findings of this study lack of focus. One of the major findings - the theme "A kinder and good assisted birth" described 'the Odon Device was kinder than other devices and described the births as 'softer', 'nicer', 'beautiful', 'easy', 'smooth' and 'gentle'. But why and how it differed from other devices needs more explanation. Furthermore, the experience of a novel device is expected to include more aspects. They could be about the reflection of the device use, e.g. convenience, impact on women and neonates emotionally and
---

	physically e.g. pain, discomfortableness, etc. 9. Some themes seem not specific to the Odon Device. For example, 'Initial altruism overtaken by genuine engagement and interest', 'Everyone's situations are different' are talking about the childbirth process. 10. The conclusion is more relevant to the study methodology, not relevant to the device itself.
--	---

VERSION 1 – AUTHOR RESPONSE

REVIEWER 1, OVERALL COMMENT: Many thanks for giving me this opportunity to review this paper that focuses on women's experiences of the Odon device when an assisted birth is required. It is a timely topic as there are growing calls to improve the use of instruments during birth, so the innovation of a new device that includes the voices of the women's experiences (and reports from the health professionals) is a vital contribution. However, the stated objectives to the findings presented are somewhat disjointed. Most of the findings related to the recruitment processes, motivations and experience of participating in such a trial. This is really important and valuable information but would need substantial revision to ensure there is consistency throughout the paper. For example, the first half of the paper indicates insights about the experience of an Odon device birth, yet only one theme that appears to be abbreviated into a Table reflects those experiences. The rest of the findings, discussion and conclusion relates to the successful PPI engagement and recruitment strategies rather than the birth experience itself. As stated, I feel this is a tremendous success but the paper needs revising accordingly to ensure the key messages are consistent and maintained throughout the paper.

AUTHOR RESPONSE: We would like to thank the reviewer for their insightful comments and positive summary. We hope we have incorporated their suggestions to their satisfaction.

Regarding the comment on insights about the experience of Odon birth: we have amended the title to be more representative of the paper – which comments on both the insights of the Odon birth but also participation in intrapartum research.

REVIEWER 1, ABSTRACT OVERALL COMMENT: Abstract: Clear, concise and well written (however, will need amending following revisions).

AUTHOR RESPONSE: Thank you for your comments, the abstract has been amended to incorporate your comments on the manuscript.

REVIEWER 1, INTRODUCTION COMMENT 1: I suggest adding a bit more contextual info i.e. reasons as to why AVB use has declined, the concerns around the other instruments (morbidity) etc, to further contextualise/funnel into the information about the Odon device.

AUTHOR RESPONSE: Thank you for this comment. We have added further contextual information into the introduction as per your suggestion, and introduction comment 2 below asking for more of the word count to be dedicated to unpacking the issues surrounding AVB. It was originally missing from the manuscript draft due to word count limitations.

Declining rates of AVB have been attributed to concerns regarding: complications of AVB and the associated litigation,⁵ limited availability of devices in certain settings, public perception,⁶ and insufficient training in device use or complex interventions.⁷

AVB is associated with better maternal and neonatal outcomes than Caesarean section at full dilatation, when performed by a skilled operator in an appropriate setting.⁸ Specifically, women having an AVB have reduced rate of postpartum haemorrhage and reduced length of stay, and babies are less likely to be admitted to the neonatal intensive care unit (NICU).⁸

AVB is almost always performed in an emergency setting, limiting time for information provision.⁹ Nevertheless, it is important to ensure that women and their birth partners, feel informed, empowered and supported during their birth experience.^{10,11} A recent review suggests that AVBs can be seen as positive by women if they receive good communication from the team and feel involved in decision-making.¹² This is despite the general negative public perception of AVB, particularly forceps. The consequences of a poorly performed AVB can be devastating and long lasting for mother, baby and operator. Comprehensive training, appropriate supervision and support can drastically improve outcomes and patient and operator experience.¹³⁻¹⁶

Page 5-6. Line 90-106

REVIEWER 1, INTRODUCTION, COMMENT 2: I'm also not convinced such a lengthy point about the justification for the case study approach is needed in the Intro, rather integrate the justifications into the methods under a sub-heading 'research design' and use the word count to unpack the issues around AVB more.

AUTHOR RESPONSE: We thank you for this comment. We added in the justification for case study methodology as it had previously been questioned. However, we have now streamlined the section (deleted a paragraph) in the introduction and will focus on this more purely in the methods with the suggested subheadings you have recommended below. Case study introduction streamlined

Moved to Page 7. Line 132-139

REVIEWER 1, INTRODUCTION, COMMENT 3: Given my comments under results, this may be better revised to include information of challenges/successes of intervention trials particularly in maternity care, particularly during 2nd stage of labour

AUTHOR RESPONSE: We thank you for this comment and your observation. We have added further information in the introduction to reflect this.

There is limited research exploring the experiences of women regarding recruitment processes during intrapartum research²⁶, those that exist report that gaining consent for intrapartum research can be challenging, especially if women are in pain or fatigued.²⁷

Page 6. Line 121-124

REVIEWER 1, METHODS, COMMENT 1: Re-ordering for a better flow and with use of more subheadings to guide the reader (I had to go back and forth to fully follow this section) e.g. Research design, study context (hospital info), recruitment & sampling and so forth. Also type of data collected be moved to the 'data collection' section. And would suggest the PPI section comes before data collection/analysis, otherwise leaves the reader wondering about the ethical considerations of the study.

AUTHOR RESPONSE: Thank you for this comment and suggestion. We have completely re-structured the methods section adding in subheadings to improve readability. We now have the subheadings: research design, study context, data sources, participants, sampling, PPI, data collection, triangulation of data, data analysis and reflexivity. We have not highlighted the changes below as we think they will be hard to follow with sporadic scored and highlighted text.

Page 7-9. Line 128-191

REVIEWER 1, METHODS, COMMENT 2: Would suggest giving some examples of the interview questions or at least the topics covered as there is only one mention (abstract) that the study was to explore motivation/participation in the study itself in addition to the experience of an Odon device birth.

AUTHOR RESPONSE: Thank you for this comment. We have added the topic guide for interviews with women as a supplementary file (Supplementary file 2)

Page 10. Line 202

REVIEWER 1, METHODS, COMMENT 3: Data analysis needs some further explaining and justification as to the methods used. Which method was followed and according to what framework/guidance/process? The references include Braun & Clarke which indicates thematic analysis and then Brink et al for grounded theory, be good to understand the rationale. If methods are combined or a novel approach has been taken, then this needs to be explicit. Great to see a reflexivity statement!

AUTHOR RESPONSE: Thank you for this comment and sorry for leaving out the detail, this was an oversight. We have now added this information.

Thematic analysis was used for analysis, using the key steps that have been well described in the literature.³⁵

Page 10. Line 218-219

Finally, case summaries were developed and reviewed to create thematic accounts which identified patterns within the data and provided explanations for these.³⁴

Page 11. Line 224-225

REVIEWER 1, RESULTS, COMMENT 1: Info about data saturation needs to be in methods (and again suggests GT approach not TA, as data saturation is highly contested in other methodologies, needs revising please). Keep results to what you have found, again, makes it easier for the reader to follow and enhances clarity.

AUTHOR RESPONSE: Thank you. We have removed the data saturation comment from the results section.

Observations ceased when no new insights were derived from two consecutive observations.

Page 9. Line 194-195

REVIEWER 1, RESULTS, COMMENT 2: Info about the observations/interviews (see lines 218-222) also would read better in the methods section for the same reasons.

AUTHOR RESPONSE: Thank you for this comment. We have added this information into the methods section.

In the methods:

Each operator and/or midwife could be interviewed more than once during the course of the study as each case was defined by the use of the Odon Device and not the clinicians present. Where possible, all interviews were undertaken face-to-face.

In the results:

Observations varied in length from 33 to 68 minutes. Interviews with women lasted 7 to 10 minutes, ~~all were undertaken face-to-face the day after the AVB.~~ interviews with midwives lasted 3 to 13 minutes and ~~A total of 11 midwives were interviewed, nine clinical midwives and two research midwives. Each midwife was interviewed at least once with some undertaking up to four interviews.~~ interviews with operators lasted between 5 and 26 minutes.

Page 10. Line 203-206

REVIEWER 1, RESULTS, COMMENT 3: The themes address women's motivation to participate and their experiences, but it appears 3 out of 4 themes are about recruitment/motivation to participate, this needs clarification throughout the paper from the abstract on and consider adapting the title and amending the figure. They are really important findings and great contribution to the field, but it reads as disjointed to the stated objectives where I was expecting much more on the experience of an Odon birth.

AUTHOR RESPONSE: We thank you for this comment. As addressed above the title has been changed to reflect this as has the objective which now reads as follows:

This study aimed to investigate women's experiences of the Odon Device and participation in intrapartum research through exploring women's and health professional views and combining them with observations collected as part of a broader case study investigation embedded within a single arm feasibility study of the Odon Device for clinically indicated AVB – the ASSIST Study

Page 7. Line 125-129

REVIEWER 1, RESULTS, COMMENT 4: Theme 2, the quote used at first glance reads as coercive, I had to read it a couple of times to realise the point, I would suggest changing the title or using a different theme name (a quote wasn't used in Theme 1). It also suddenly introduces key information about the recruitment strategy not evident in the methods. The use of videos is great but is essential to know about it in the methods.

AUTHOR RESPONSE: Thank you for this comment. The recruitment information was in the methods however, as you have previously stated, was perhaps a little lost with the lack of signposting to the reader. The recruitment methods should now be much clearer as it has its own subtitle in the methods section. We have also taken on board your comment about the title of theme 2 and have amended this.

Theme two: ~~'That woman that come around, she had the right approach'~~ What makes an acceptable recruitment process?²

Page 18. Line 369

REVIEWER 1, RESULTS, COMMENT 5: Theme 3 is underdeveloped and doesn't provide very rich insights to the experiences. The use of a table without the narrative doesn't sit in line with the other themes.

AUTHOR RESPONSE: We appreciate your comment and purely inserted a table, without a detailed narrative, due to the limitations of the word count. We have now changed this by removing the table and providing a narrative report. Changes are not detailed here as they are extensive, and can be found under theme three.

Page 19-21. Line 396-436

REVIEWER 1, RESULTS, COMMENT 6: Theme 4 doesn't have enough insights to be a theme, would suggest integrating it into another.

AUTHOR RESPONSE: Thank you for your comment. We have integrated it into theme 1.

No change to text (apart from deleting the heading of theme 4)

REVIEWER 1, DISCUSSION, COMMENT 7: Given my other comments I would suggest revising this section, needs greater exploration of the wider literature including women's views/experiences of AVB to support this section. Nor does it link to operators views and experiences which would strengthen the points made here.

AUTHOR RESPONSE: Thank you for your comment. We have revised this section. We have not linked the discussion to the operator/midwife views or experiences as this is not the focus of the manuscript and we would be discussing data that we have not presented in the results section.

These findings are very similar to a recent systematic review on women's experiences of AVB.¹² As it was the first time the Odon Device had been used in clinically indicated cases²², there was acceptance and expectation from women that their births be formally observed. There was less evidence that video-recording the births was acceptable to women.

Page 22. Line 484-488

REVIEWER 1, CONCLUSION, COMMENT 1: Reads well and marries up to my comments about the key point of this paper which appears to be about successful PPI and recruitment engagement to a challenging field for intervention trials. If the manuscript is revised with this conclusion in mind, it will provide a structured 'golden thread' with valuable information for other healthcare professionals and trialists.

AUTHOR RESPONSE: We thank the reviewer for this kind comment and we hope we have reflected their suggestions in the changes to the manuscript.

REVIEWER 2, OVERALL COMMENT: The study aimed to investigate experiences of women who gave birth assisted by the Odon Device, an innovative device for assisted vaginal birth via observation and individual interviews with women, women's partners, operators, and midwives. The research topic is interesting and the manuscript provides some useful information for the practice of Odon Device for assisted childbirth. However more details of the reflections and perspectives about the experience of device usage are expected and it will contribute to the application of Odon Device in obstetric practice and the improvement of clinical service quality.

AUTHOR RESPONSE: We thank the reviewer for their overall comment and hope we have incorporated their suggestions to their satisfaction. Data relating to optimising use of the device by operators has been analysed separately and is being prepared as a manuscript to submit to BMJ Open.

REVIEWER 2, COMMENT 1: Authors described this study used a case study research method. However, the observation and individual interviews are different kinds of qualitative study. The research methods used in this study, including sampling, data collection and data analysis, seem more like qualitative research methods. Please reconsider the methodology of this study.

AUTHOR RESPONSE: We thank the reviewer for their comment. We can confirm we used case study methodology, which combines a number of qualitative data sources to illuminate each case and also involves cross case comparison. The qualitative data collection methods used in the case study methodology were observations and interviews. We hope to have clarified this better in the methods section, incorporating comments from reviewer 1 as well.

Full changes not listed here as the methods have now been extensively revised.

Page 7-11. Line 131-240

REVIEWER 2, COMMENT 2: It is expected to introduce the Odon Device in Background in more details such as the conditions of application, approaches of its usage, and the innovation and difference compared to other devices.

AUTHOR RESPONSE: We thank you for this comment. These details have already been published in full in our manuscripts referenced in this paper (Reference 22 and 28). We reference this work in the final paragraph of the Introduction.

REVIEWER 2, COMMENT 3: For qualitative data analysis, it is usually required two independent researchers to carry out coding. This study reported 'Double coding of a proportion of interview transcripts was undertaken by JW'. Please provide the specific proportion in this section.

AUTHOR RESPONSE: Thank you for this comment. We have added the proportion that was double coded into the text.

Double coding of a proportion (20%) of interview transcripts was undertaken by JW.

Page 11. Line 226

REVIEWER 2, COMMENT 4: Table 2 seems not necessary and its contents can be described briefly. A table introducing the characteristics of participants might be needed.

AUTHOR RESPONSE: Thank you for this comment. We believe that given the lack of familiarity of many readers with qualitative case study methodology table 2 is important in allowing transparent presentation of the data collected for each case.

REVIEWER 2, COMMENT 5: Most of the first three paragraphs of Result are about data collection. It is supposed to be in Methodology part.

AUTHOR RESPONSE: We thank you for this comment (which has also been noted by reviewer 1). We have made changes to the beginning of the results section that address this issue.

REVIEWER 2, COMMENT 6: For some quotes, it is a bit hard to catch. For example, in P16, --M11: 'And then at that point she said, "Oh, come on doctor," she said something like, "I've got faith in you, you can do it." So, she really, really wanted him to do it. And then she said, "But if this doesn't work, I don't want forceps because of the baby's face; I'd rather have a Section.'" It is confusing whom 'she' and 'him' referred to. Maybe 'She' was woman, and 'him' was the doctor.

AUTHOR RESPONSE: Thank you for this observation. We have added clarification into the quote.

M11: *'And then at that point she said, "Oh, come on doctor," she said something like, "I've got faith in you, you can do it." So, she [the patient] really, really wanted him [the operator] to do it. And then she [the patient] said, "But if this doesn't work, I don't want forceps because of the baby's face; I'd rather have a Section."*

Page 16. Line 338-339

REVIEWER 2, COMMENT 7: P19, it is suggested to use "pregnancy outcomes or complications" rather than "birth outcomes or complications" in the sentence "women focused on outcomes for their baby and their feelings at the time of birth rather than their own birth outcomes or complications."

AUTHOR RESPONSE: Thank you for this comment. We have amended the text to reflect your suggestion.

Interestingly, women focussed on outcomes for their baby and their feelings at the time of birth rather than their own ~~birth~~ pregnancy outcomes or birth complications when discussing their experiences.

Page 19. Line 404-405.

REVIEWER 2, COMMENT 8: The findings of this study lack of focus. One of the major findings -the theme "A kinder and good assisted birth" described 'the Odon Device was kinder than other devices and described the births as 'softer', 'nicer', 'beautiful', 'easy', 'smooth' and 'gentle'. But why and how it differed from other devices needs more explanation. Furthermore, the experience of a novel device is expected to include more aspects. They could be about the reflection of the device use, e.g. convenience, impact on women and neonates emotionally and physically e.g. pain, discomfortableness, etc.

AUTHOR RESPONSE: We agree with the reviewer comments. Comprehensive assessment of novel devices is of paramount importance. However, this manuscript is focussed purely on the experiences of women and data relevant to optimising use of the device is in progress to submit as a separate manuscript to BMJ Open. The impact on women and neonates (quantitative data) has been published already and referenced within this manuscript. We have added a further quote from a woman supporting the belief that the birth was positive due to minimal birth trauma visible to their baby.

W07: *'Over the moon. The fact that it has helped with his delivery, the fact that he's safe, there's no marks on him whatsoever, he's completely fine, and he's just a happy baby. So, that's all I could ever ask for really.'*

Page 15. Line 323-325

REVIEWER 2, COMMENT 9: Some themes seem not specific to the Odon Device. For example, 'Initial altruism overtaken by genuine engagement and interest', 'Everyone's situations are different' are talking about the childbirth process.

AUTHOR RESPONSE: Thank you for this comment. An oversight of the first draft was that we failed to highlight clearly that this study focussed on the experiences of women both on the use of the Odon device but also on the recruitment to intrapartum research. We hope that the amendments we have made to manuscript reflect this.

REVIEWER 2, COMMENT 10: The conclusion is more relevant to the study methodology, not relevant to the device itself.

AUTHOR RESPONSE: We thank you for your comment and have amended the conclusion to be more related to the device.

When introducing or implementing a novel device into clinical use it is vital that the experiences and opinions of stakeholders are considered. Case studies provide a way of exploring these views in detail by integrating data from multiple sources, providing a comprehensive and authentic exploration. Case study methodology is used widely in other avenues of research however it has not yet been adopted in implementation research during device or surgical trials. Successful completion of case studies on labour ward for an intrapartum intervention has provided methodology that can be transferred to intervention delivery and design of novel devices. Women appear keen to support the investigation of novel devices for AVB, such as the Odon Device, and value the importance of information sharing during the recruitment process. ~~Furthermore, this methodology could be used within the broader context of randomised controlled trials for novel devices both within obstetrics and beyond. They have the potential to inform not only the conduct of trials but also provide insight into how stakeholders perceive an intervention.~~ These perceptions of women can help shape and design future recruitment strategies, information provision and intervention delivery. The use of case study methodology must be further tested in alternative settings and specialties.

Page 25. Line 544-549.

VERSION 2 – REVIEW

REVIEWER	Claire Feeley University of Central Lancashire
REVIEW RETURNED	15-Nov-2021

GENERAL COMMENTS	Great to see this revised paper and I appreciate the work the authors have gone to in order to address reviewer comments. This paper reads with much greater clarity and focus. I have no further comments.
---

REVIEWER	Hong Jiang Fudan University
REVIEW RETURNED	21-Nov-2021

GENERAL COMMENTS

The comments were well addressed by authors. I have no further comments.